# Micromanufactured Tactile Samples for Characterization of Rough and Dry Tactile Perception

**DOI:** 10.3390/mi13101685

**Published:** 2022-10-07

**Authors:** Keiichiro Yanagibashi, Norihisa Miki

**Affiliations:** Department of Mechanical Engineering, Keio University, Yokohama 223-8522, Kanagawa, Japan

**Keywords:** haptics, microfabrication, micromanufacturing, tactile perception, tactile sample, roughness, dry, wet, tactile dimensions

## Abstract

The quantitative characterization of tactile perception, which is crucial in the design of tactile devices, requires the tested samples to have individually and precisely controlled properties associated with the senses. In this work, we microfabricated such tactile samples and then quantitatively characterized tactile perception with a focus on roughness and dryness. In the roughness perception experiments, the tactile samples had a stripe pattern with ridge and groove widths that were individually controlled. The experimental results revealed that the feeling of roughness was more dominated by the width of the groove than that of the ridge and that conventionally used roughness parameters such as $S_{a}$ and $S_{q}$ were not sufficient for predicting roughness perception. In the dryness perception experiments, the tactile samples had a micropattern formed by dry etching and an array of squares. The experimental results revealed that dry perception had different properties when the feature sizes were below and above 30 µm, which may have been due to the effect of adhesion on friction. The proposed tactile samples were suitable for the quantitative and precise characterization of tactile perception.

## 1. Introduction

The tactile perception of surface textures has been studied by many research groups. It is categorized into several psychological tactile dimensions, namely roughness (rough/smooth), stiffness (hard/soft), moisture (moist/dry), friction (sticky/slippery), and warmness (warm/cold) [1,2]. In studies, participants are requested to touch a surface texture and report their feeling using methods such as the semantic differential method, the paired comparison method, and the rating method [3,4]. The results are statistically analyzed with respect to the physical properties of the surface textures. One challenge in such studies is that the physical properties of the textures are difficult to independently control. For example, in our prior study, we used 18 test samples that included wood, polystyrene form A, polystyrene form B, urethane, lumpy rubber, and flat rubber [5,6]. Each sample had different surface properties with respect to parameters such as geometry, roughness, stiffness, friction coefficient, surface energy, and thermal conductivity. To correlate the tactile perception to the individual physical properties, the test samples should have identical properties except for the physical property of interest. For example, when evaluating stiffness, it is preferable to test samples that have different stiffness values but are otherwise identical. Micromanufacturing technologies can be used to develop such tactile samples. For example, photolithography or machining can be used to prepare surfaces with textures on the micrometer to millimeter scale for a given material. An identical surface geometry can be patterned onto different materials using micromolding, photolithography, or machining. Various tactile samples have been reported in the literature, including 3D-printed textures with cone-shaped textons [7], metal samples patterned with picosecond laser pulses [8], photolithographically micromanufactured silicon [9] and polymer samples [10], and stretchable tactile samples with micropatterned features that can be varied [11].

In this study, using tactile samples, we quantitatively characterized the perceptions of roughness and dryness of surface textures and correlated the results with the physical properties of the surfaces. For the roughness perception experiments, we prepared tactile samples with a stripe pattern with ridge and groove widths that were independently controlled. The effect of the pattern dimensions on perception was experimentally characterized. For the dryness perception experiments, we prepared surfaces with a random pattern or a square pattern. Using tactile samples, we experimentally determined the most dominant parameter that affected dryness. The perception experiments were approved by the Research Ethics Committee of the Faculty of Science and Technology, Keio University (31-52).

## 2. Tactile Samples and Methods

### 2.1. Ranking Method

In this work, the ranking method was used to investigate the relationship between the surface properties and tactile perception. First, we prepared the tactile samples with surface textures that were precisely controlled by microfabrication. Then, the participants were requested to rank the samples with respect to the tactile perception of interest such as roughness or dryness. The null hypothesis was the independence of the rankings given by the participants. If the null hypothesis was rejected, we could conclude that the tactile perception was affected by the textures.

Kendall’s coefficient of concordance W was obtained using the following equations [12]:(1)$$R=\frac{n\left(k+1\right)}{2}$$
(2)$$S=\sum {\left(R_{i}-R\right)}^{2}$$
(3)$$W=\frac{12S}{n^{2}\left(k^{3}-k\right)}\;$$
where n and k are the numbers of participants and samples, respectively, and i=1…k.

The null hypothesis was the independence of the rankings given by the participants. A chi-square test was conducted.
(4)$$\chi_{r}{}^{2}=n\left(k-1\right)W$$

$\chi_{r}{}^{2}$ was compared with $\chi^{2}{}_{\left(k-1,\;p\right)}$ at the degree of freedom of (k−1) and the significance level α. When k=10 and α=0.05, $\chi^{2}{}_{\left(9,0.05\right)}=16.92$.

### 2.2. Roughness Perception Experiments

#### 2.2.1. Tactile Samples for Roughness Perception Experiments

Roughness perception is classified into macroroughness and microroughness perception [13] based on the responsible tactile receptors. Macroroughness, for which the corresponding surface pattern is greater than several hundreds of micrometers, is perceived via Merkel’s disks [14,15,16]. Microroughness, for which the corresponding surface pattern is smaller than 1 mm, is perceived via Meissner corpuscles and Pacinian corpuscles [17,18,19]. Surface-texture patterns ranging from 0.1 to 1 mm are considered to have the properties of both macro- and microroughness and are thus of great interest in roughness perception studies.

To prepare our samples, a negative photoresist (SU-8 3050, Kayaku Advanced Materials, Westborough, MA, USA) was patterned onto a glass substrate. The thickness of the SU-8 was 50 µm, which was sufficient for perception. Stripe patterns with a ridge width w and a groove width p were patterned using photolithography (see Figure 1). In this work, the effects of w and p on roughness perception were investigated. Tactile samples with w varying from 0.1 to 1 mm in 0.1 mm steps and p fixed at 0.5 mm (p-constant samples) and those p varying from 0.1 to 1 mm in 0.1 mm steps and w fixed at 0.5 mm (w-constant samples) were prepared.

#### 2.2.2. Roughness Perception Experiments with Tactile Samples

The effects of surface textures, the ridge width w, and the groove width p were experimentally investigated; w and p were precisely controlled. The ranking methods were conducted with 10 w-constant samples and 10 p-constant samples using 11 participants (10 males and 1 female, aged 20 to 29 years). The participants were requested to not look at the tactile samples during the tests. The participants ranked the 10 samples from 1 (least rough) to 10 (most rough).

Next, in order to investigate which parameter was more dominant (w or p), 10 samples composed of 5 p-constant ((w, p) = (0.2, 0.5), (0.4, 0.5), (0.6, 0.5), (0.8, 0.5), and (1.0, 0.5)) and 5 w-constant samples ((w, p) = (0.5, 0.2), (0.5, 0.4), (0.5, 0.6), (0.5, 0.8), and (0.5, 1.0)) were ranked by 5 participants (5 males, aged 20 to 29 years). The participants conducted the ranking experiments with the 10 samples twice.

### 2.3. Dry/Wet Perception Experiments

#### 2.3.1. Tactile Samples for Dry/Wet Perception Experiments

Two types of tactile samples were prepared, namely those with random and square patterns. The samples with a random pattern were composed of polydimethyl siloxane (PDMS). The surface patterns were formed using dry etching with CF_4_ and O_2_ plasma [20,21]. The PDMS was prepared with a mixing ratio of the base to the curing agent of 10 (SYLGARD 184 W/C, DOW CORNING TORAY Co., Tokyo, Japan). The dry etching was conducted using an etcher (FA-1, Samco Inc., Kyoto, Japan) with a radio frequency power of 150 W and a flow rate ratio of CF_4_ to O_2_ of 3 to 1 at a chamber pressure of 0.1 MPa. The etching time was varied from 10 to 300 min. The surface roughness $S_{a}$ was measured with a laser microscope (VK-X100, Keyence Co. Ltd., Osaka, Japan). Figure 2 shows photographs of the samples treated with dry etching for 30, 60, 120, and 240 min.

The square pattern was formed with a negative photoresist (SU-8 3005, Kayaku Advanced Materials, Inc., Westborough, MA, USA). The width of the square w and the pitch or the gap between squares p were varied from 10 to 15 µm in steps of 5 µm. When w=p, the ratio of the top surface to the etched surface was constant.

#### 2.3.2. Dry/Wet Perception Experiments Using Tactile Samples with Random Patterns

The ranking method was conducted to investigate the effects of surface roughness on dry/wet perception using the tactile samples with a random pattern. A total of 9 participants (8 males and 1 female, aged 20 to 29 years) ranked the samples.

#### 2.3.3. Dry/Wet Perception Experiments Using Tactile Samples with Square Patterns

In order to investigate the effects of the surface textures on dry/wet perception, the ranking method was conducted using samples with the same w and p with respect to the dry feeling. A total of 14 participants (12 males and 2 females, aged 20 to 29 years) ranked the samples. As will be described later, two different trends were found for the feature sizes below and above 30 µm.

To investigate this trend, samples with a width w of 15 or 25 µm (below 30 µm) were compared in terms of a dry feeling while p was varied from 10 to 50 µm. Samples with a w value of 40 or 50 µm (above 30 µm) were then compared. The number of participants was 7 (all males, aged 20 to 29 years). Each participant conducted the comparison twice per condition.

Finally, the effect of the pitch p was investigated for samples with a w value of 15, 25, 40, or 50 µm while p was 5, 10, 15, 20, 25, 30, 35, 40, 45, or 50 µm. The number of participants was 10 (9 males and 1 female, aged 20 to 29 years).

All the experiments described in Section 2 are summarized in Table 1.

**Table 1 micromachines-13-01685-t001:** Summary of the experiments.

Tactile Perception	Texture of Tactile Samples	Parameters	Objectives: To Investigate	Number of Participants	Results
Roughness	Stripe (Figure 1)	Ridge and groove widths w, p	The effects of the ridge and groove widths	11 (10 males and 1 female, aged 20 to 29 years)	Section 3.1
			Which of the ridge and the groove widths was more dominant	5 (5 males, aged 20 to 29 years)	Section 3.1
Dryness	Random (Figure 2)	Etching time/roughness $S_{a}$	The effects of the surface roughness	9 (8 males and 1 female, aged 20 to 29 years)	Section 3.2.1
	Square (Figure 3)	Square width w and gap between the squares p (w=p)	The effects of the feature size	14 (12 males and 2 females, aged 20 to 29 years)	Section 3.2.2
		Square width w and gap between the squares p	How the dryness perception varied with the feature size below and above 30 µm	7 (7 males, aged 20 to 29 years)	Section 3.2.2
			The effects of the gap	10 (9 males and 1 female, aged 20 to 29 years)	Section 3.2.2

## 3. Results and Discussion

### 3.1. Roughness Perception Experiments

Figure 4a–c shows the ranking for the 10 w-constant samples, 10 p-constant samples, and a mix of w- and p-constant samples, respectively. The relationship shown in Figure 4a was opposite to that shown in Figure 4b.

For the w-constant samples, $\chi_{r}{}^{2}=81.9$ and $\chi_{r}{}^{2}>\chi^{2}{}_{\left(9,0.05\right)}$. For the p-constant samples, $\chi_{r}{}^{2}=63.5$ and $\chi_{r}{}^{2}>\chi^{2}{}_{\left(9,0.05\right)}$. Therefore, for both cases, the rankings were concordant.

Figure 4c shows that a larger pitch p led to a larger roughness feeling and that p was more dominant over w. For this mix of samples, $\chi_{r}{}^{2}=83.8$ and $\chi_{r}{}^{2}>\chi^{2}{}_{\left(9,0.05\right)}$. This result showed that for a stripe pattern with a width and a pitch from 0.1 to 1.0 mm, the gap was the dominant factor that affected the feeling of roughness.

### 3.2. Dry/Wet Perception Experiments

#### 3.2.1. Experiments with Tactile Samples with Random Patterns

The ranking method was conducted for 15 samples. The dry feeling increased with the sample etching time, as shown in Figure 5. The concordance of the ranking was confirmed via chi-square tests, where $\chi_{r}{}^{2}=107.7>\chi^{2}{}_{\left(14,0.05\right)}=23.7$.

Since the roughness $S_{a}$ increased with the sample etching time, a correlation between the dry feeling and $S_{a}$ was obtained (see Figure 6a,b). However, as discussed in Section 3.3, $S_{a}$ did not identify the surface geometry, which had to be carefully investigated using tactile samples.

#### 3.2.2. Experiments with Tactile Samples with Square Patterns

Figure 7 shows the ranking of the dry tactile feeling for samples with an equal square width w and gap p. For these samples, $\chi_{r}{}^{2}=41.7>\chi^{2}{}_{\left(8,0.05\right)}=15.5$ and the statistical concordance was validated. Interestingly, two trends were observed: one for w≤25 and the other for w≥35, with a transition at w=30. When w≤25, the dry tactile feeling increased with w. However, when w≥35, w did not affect the dry tactile feeling. For the four samples with w≥35, $\chi_{r}{}^{2}=0.343<\chi^{2}{}_{\left(3,0.05\right)}=7.815$ and no statistically significant difference was found. This result implied that the perception mechanism of the dry/wet feeling was different when the surface features were larger or smaller than 30 µm.

Next, samples with w values of 15 and 25 µm and 40 and 50 µm were compared for pitches p of 10, 20, 30, 40, and 50 µm. Seven participants (seven males, aged 20 to 29 years) conducted the perception tests twice (the number of trials was 7×2=14). When 11 or more trials showed the same results, the test was statistically significant at a significance level of 0.05. Table 2 shows the comparison results for samples with w values of 15 and 25 µm. In the table, “+” indicates that a line (w=25) produced a drier feeling than a row (w=15) and that this difference was statistically significant; N.S. means not significant; and “− “ indicates that the line produced a less dry feeling than the row. The results suggested that a larger p resulted in a drier feeling.

Table 3 shows the comparison results for samples with w values of 40 and 50 µm. The dry feeling depended on the pitch p. However, a trend different from that above was found. For samples with a given p value, w values of 15 and 25 µm did not produce a statistically significant difference. For samples with w values of 40 and 50 µm, significant differences were found when p was 30, 40, and 50 µm. Of note, a w value of 40 µm produced a rougher feeling than did a w value of 50 µm, as summarized in Table 4. This different trend may have originated from the perception properties shown in Figure 7.

The dry feeling was investigated for samples with a given w value. Figure 8a–d show the results for w values of 15, 25, 40, and 45 µm, respectively, for various pitch p values. A total of 10 participants (9 males and 1 female, aged 20 to 29 years) conducted the ranking experiments. For these samples, $\chi^{2}{}_{\left(9,0.05\right)}=16.9$; $\chi_{r}{}^{2}=61.4$, 67.6, 71.8, and 67.8, respectively, and for the cases; and $\chi_{r}{}^{2}>\chi^{2}{}_{\left(9,0.05\right)}$. A drier feeling was produced with an increasing pitch p.

When the rankings of samples with p values of 35, 40, 45, and 50 µm were extracted, $\chi_{r}{}^{2}=5.46$, 4.2, 2.16, and 12.4 for w values of 15, 25, 40, and 45 µm, respectively; and $\chi^{2}{}_{\left(3,0.05\right)}=7.815$. Except for the case with a w value of 50 µm, no statistically significant difference in the rankings was found. This result was consistent with that shown in Figure 7 and agreed with the idea that the perception mechanism of the dry/wet feeling was different between feature sizes below and above 30 µm.

### 3.3. Discussion

#### 3.3.1. Roughness Feeling

The roughness feeling increased with the width w and decreased with the pitch p. Surface roughness is conventionally evaluated using the arithmetic mean height $S_{a}$, the root mean square roughness $S_{q}$, or both. These parameters are respectively expressed as:(5)$$S_{a}=\frac{1}{l_{r}}\int_{0}^{l_{r}}\left|Z\left(x\right)\right|dx,$$
(6)$$S_{q}=\sqrt{\frac{1}{l_{r}}\int_{0}^{l_{r}}{\left(Z\left(x\right)\right)}^{2}dx},$$
where $l_{r}$ is the sampling length and Z(x) is the height from the average at position x. For a surface with a striped pattern with a width w and a pitch p, $S_{a}$ and $S_{q}$ in the direction perpendicular to the stripe pattern can be expressed as:(7)$$S_{a}=\frac{2wp}{{\left(w+p\right)}^{2}}h$$
(8)$$S_{q}=\frac{h}{w+p}\sqrt{wp},$$
where h is the height of the stripe pattern. Using these equations, the width w and the pitch p had the same contribution to the conventional surface roughness parameters $S_{a}$ and $S_{q}$. Conventionally, a roughness feeling is considered to increase with the surface roughness, which is conventionally assessed using $S_{a}$, $S_{q}$, or both [22]. However, our experiments with micromanufactured tactile samples revealed that the width w and the pitch p had opposite contributions to the roughness feeling compared with those for a surface with a stripe pattern. This result agreed with previously reported results [14]. Therefore, an assessment of the surface roughness using only $S_{a}$ and $S_{q}$ is insufficient for tactile perception experiments. The bearing ratio, which describes the area of the surface above a given depth, was proposed as a measure for determining the contact state in a bonding process [23]. In our case, when the bearing depth was smaller than the depth of the features, the bearing ratio (BR) could be expressed as:(9)$$BR=\frac{w^{2}}{{\left(w+p\right)}^{2}}=\frac{1}{{\left(1+\frac{p}{w}\right)}^{2}},$$

BR is a function of (p/w); it monotonically decreases with (p/w). Therefore, BR increases with p and decreases with w, which agrees with the obtained results, as shown in Figure 4. BR was considered to represent the contact and hence it was reasonable that BR had a good correlation with roughness perception. This relation will be further investigated in our future work.

For the p- and w-constant samples, we investigated the statistical significance of the differences among all ranks. The significance level α was 0.05. For the p-constant samples, no statistically significant differences were found between the samples with w values of 0.9 and 1.0 mm, as shown in Table 5. For the w-constant samples, no statistically significant differences were found between the samples with p values of 0.5 and 0.6, 0.6 and 0.7, 0.7 and 0.8, 0.8 and 0.9, 0.8 and 1.0, and 0.9 and 1.0, as summarized in Table 6. We found that the statistical significance tended to be lost when the ratio of the two variables, called the Weber ratio, was small. For the p-constant samples, no statistically significant differences were found when the ratio was 0.11 or smaller. For the w-constant samples, no statistically significant differences were found when the ratio was less than 0.25.

#### 3.3.2. Dry/Wet Feeling

The dry feeling increased with surface roughness for samples with a random pattern. As for the roughness feeling, the properties of the surface roughness were carefully investigated using tactile samples with a square pattern. We found that the perception characteristics changed at a feature size threshold of 30 µm. For dynamic touch, it has been reported that the roughness, friction, and stickiness sensations play important roles in wetness perception [24]. A previous study reported that the dry/wet feeling had the same dimension as that of friction [25]. Experiments with molded samples that had triangular grooves revealed that the coefficient of friction was small when the pitch of the surface features was in the range of 40–120 µm and that a non-textured surface and a surface with smaller features had a large coefficient of friction due to adhesion [26]. This explains the perception characteristics observed in our experiments well.

We found experimentally that the pitch contributed more to the dry feeling than did the width, which agreed with the results for the roughness feeling. As shown in Table 7, Table 8, Table 9 and Table 10, statistically significant differences were found when the ratios of the change in the pitches was greater than 0.33, 0.33, 0.21, and 0.16 with w values of 15, 25, 40, and 50 µm, respectively.

## 4. Conclusions

In this study, we micromanufactured tactile samples with surface features that were precisely controlled. The tactile samples with a stripe pattern revealed that the conventionally used surface roughness parameters, namely $S_{a}$ and $S_{q}$, did not always represent the roughness feeling. The roughness perception increased with the gap width and decreased with the ridge width, where the gap width was dominant over the ridge width. Dry perception was found to have a threshold of 30 µm in terms of feature size, below and above which perception showed a different trend. This was considered to be due to the adhesive friction that occurred when the feature size was below 30 µm. These findings were obtained in experiments using precisely micromanufactured tactile samples.

## Figures and Tables

**Figure 1 micromachines-13-01685-f001:**
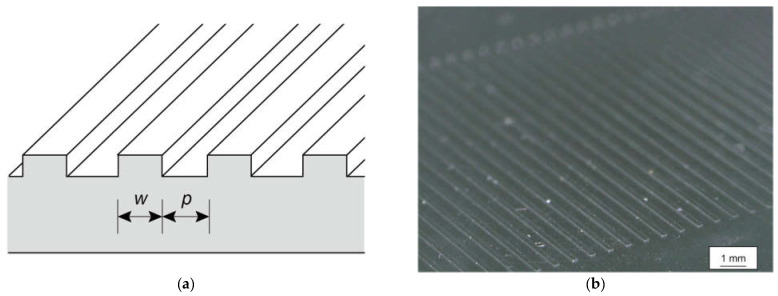
Tactile samples used for roughness perception experiments. (**a**) Cross-sectional schematic where w and p are the ridge and groove widths, respectively. The ridge height was 50 µm. (**b**) Microscopy image of sample texture.

**Figure 2 micromachines-13-01685-f002:**
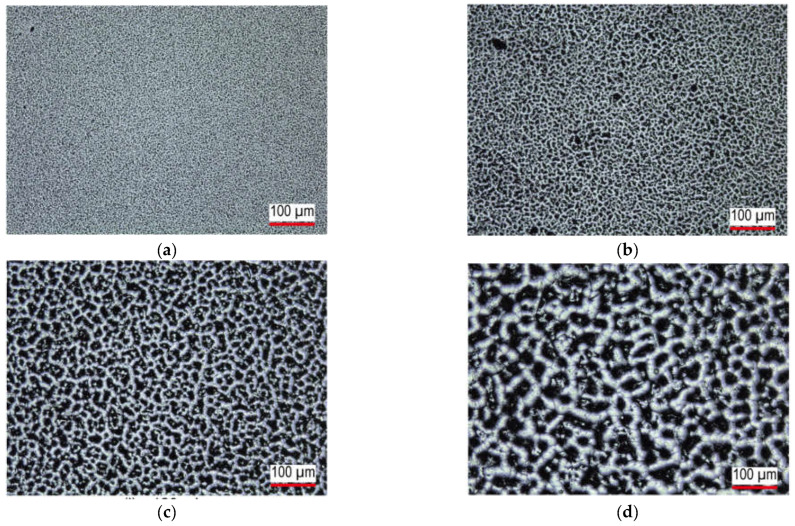
Tactile samples with random patterns for dry perception experiment. Samples treated with dry etching for (**a**) 30, (**b**) 60, (**c**) 120, and (**d**) 240 min.

**Figure 3 micromachines-13-01685-f003:**
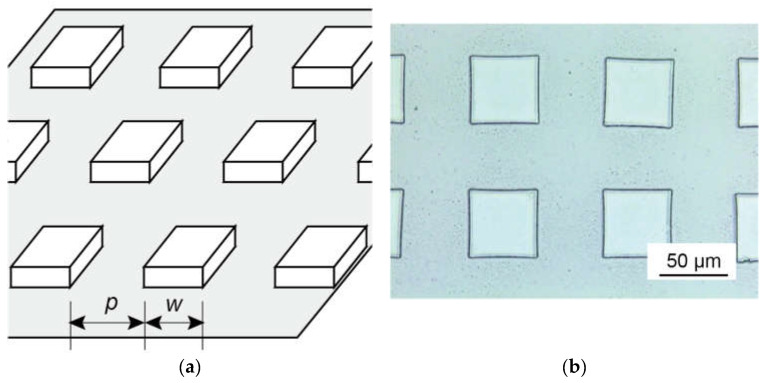
Tactile samples with square pattern for dry perception experiment: (**a**) diagram of pattern, where w is the square width and p is the gap between squares; (**b**) microscopy image of sample.

**Figure 4 micromachines-13-01685-f004:**
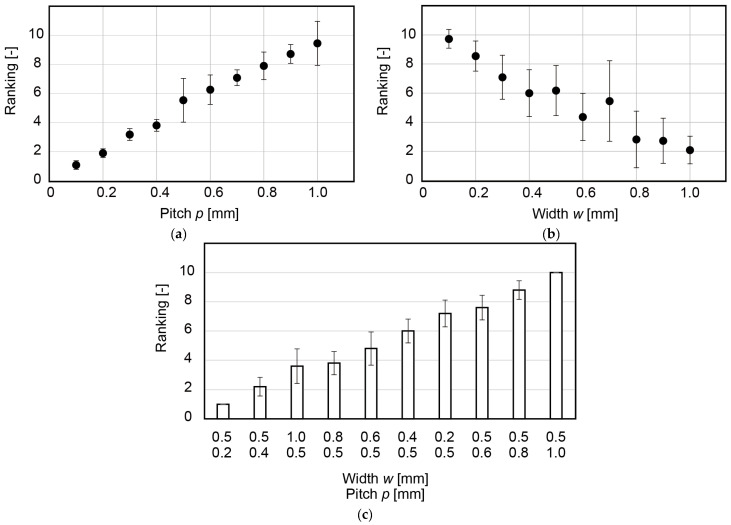
Ranking of roughness feeling for (**a**) w-constant samples, (**b**) p-constant samples, and (**c**) mix of w- and p-constant samples.

**Figure 5 micromachines-13-01685-f005:**
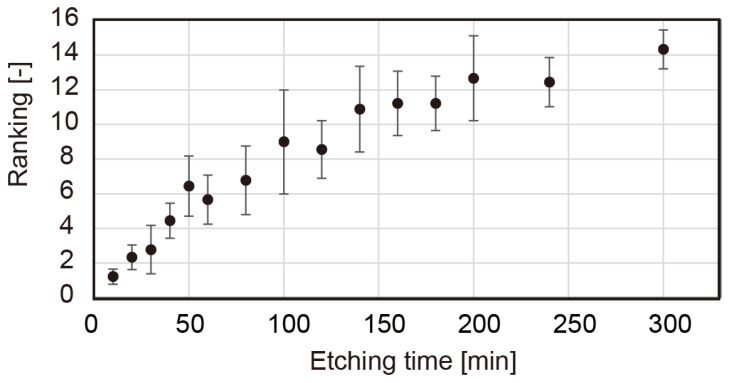
Ranking of dry perception for various etching times.

**Figure 6 micromachines-13-01685-f006:**
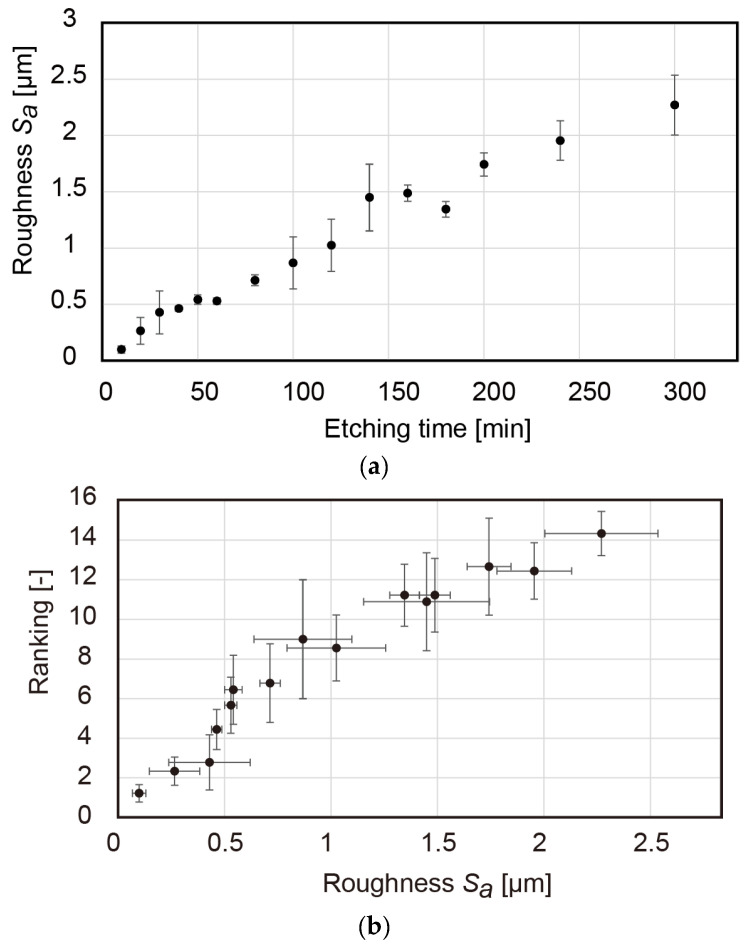
(**a**) Increase in arithmetic mean height $S_{a}$ with sample etching time; (**b**) ranking of dry perception as function of $S_{a}$.

**Figure 7 micromachines-13-01685-f007:**
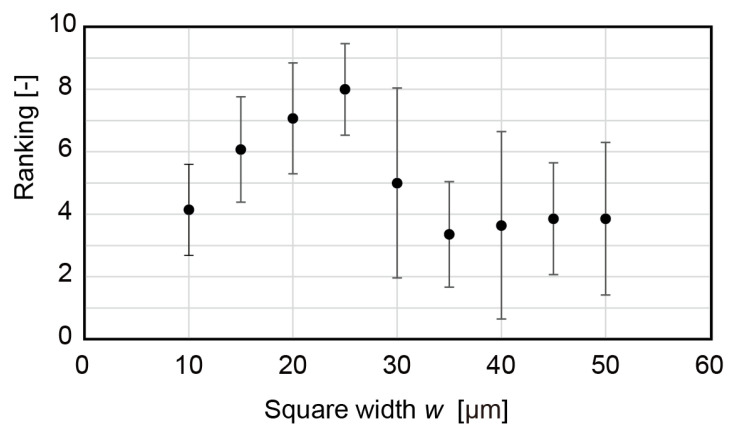
Ranking of dry perception for various square patterns.

**Figure 8 micromachines-13-01685-f008:**
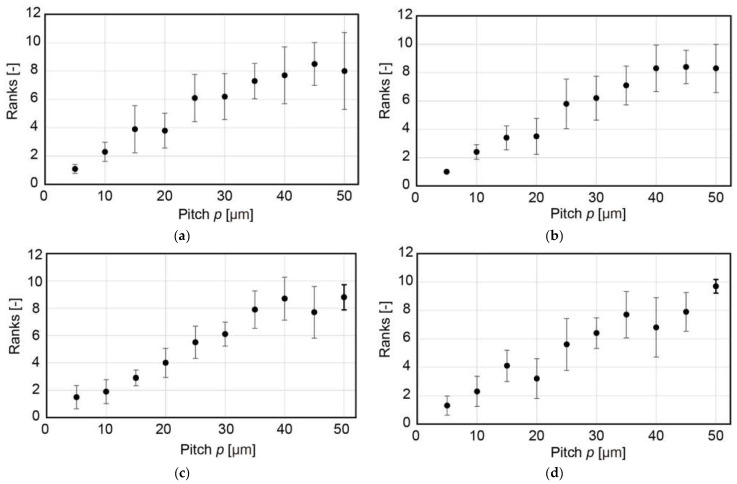
Ranking of dry perception for w values of (**a**) 15, (**b**) 25, (**c**) 40, and (**d**) 45 µm for various pitch p values.

**Table 2 micromachines-13-01685-t002:** Statistical significance in dryness perception between samples with w values of 15 and 25 µm. N.S. means not significant.

**(**w, p**) (µm)**	(25, 10)	(25, 20)	(25, 30)	(25, 40)	(25, 50)
(15, 10)	N.S.	+	+	+	+
(15, 20)	−	N.S.	+	+	+
(15, 30)	−	−	N.S.	+	+
(15, 40)	−	−	N.S.	N.S.	N.S.
(15, 50)	−	−	−	N.S.	N.S.

**Table 3 micromachines-13-01685-t003:** Statistical significance in dry perception between samples with w values of 40 and 50 µm. N.S. means not significant.

**(** w **,** p **) (µm)**	(50, 10)	(50, 20)	(50, 30)	(50, 40)	(50, 50)
(40, 10)	N.S.	+	+	+	+
(40, 20)	−	N.S.	N.S.	+	+
(40, 30)	−	−	−	N.S.	+
(40, 40)	−	−	−	−	N.S.
(40, 50)	−	−	−	−	−

**Table 4 micromachines-13-01685-t004:** Number of trials in which participants perceived that samples with a w value of 25 µm were rougher than those with a w value of 15 µm and that those with a w value of 50 µm were rougher than those with a w value of 40 µm for various p values. * Significance level < 0.05.

p **(µm)**	w: 15 –25 (µm)	w: 40 –50 (µm)
10	7	7
20	8	6
30	9	0 *
40	10	2 *
50	9	3 *

**Table 5 micromachines-13-01685-t005:** Statistical significance in roughness perception for p-constant samples with various w values. * Significance level < 0.05. N.S. means not significant.

w	0.1	0.2	0.3	0.4	0.5	0.6	0.7	0.8	0.9	1.0
0.1		*	*	*	*	*	*	*	*	*
0.2			*	*	*	*	*	*	*	*
0.3				*	*	*	*	*	*	*
0.4					*	*	*	*	*	*
0.5						*	*	*	*	*
0.6							*	*	*	*
0.7								*	*	*
0.8									*	*
0.9										N.S.
1.0										

**Table 6 micromachines-13-01685-t006:** Statistical significance in roughness perception for w-constant samples with various p values. * Significance level < 0.05. N.S. means not significant.

p	0.1	0.2	0.3	0.4	0.5	0.6	0.7	0.8	0.9	1.0
0.1		*	*	*	*	*	*	*	*	*
0.2			*	*	*	*	*	*	*	*
0.3				*	*	*	*	*	*	*
0.4					*	*	*	*	*	*
0.5						N.S.	*	*	*	*
0.6							N.S.	*	*	*
0.7								N.S.	*	*
0.8									N.S.	N.S.
0.9										N.S.
1.0										

**Table 7 micromachines-13-01685-t007:** Statistical significance in dryness perception for samples with a w value of 15 µm for various p values. * Significance level < 0.05. N.S. means not significant.

p	5	10	15	20	25	30	35	40	45	50
5		*	*	*	*	*	*	*	*	*
10			*	*	*	*	*	*	*	*
15				N.S.	*	*	*	*	*	*
20					N.S.	*	*	*	*	*
25						N.S.	*	*	*	*
30							N.S.	*	*	*
35								N.S.	N.S.	*
40									N.S.	N.S.
45										N.S.
50										

**Table 8 micromachines-13-01685-t008:** Statistical significance in dryness perception for samples with a w value of 25 µm for various p values. * Significance level < 0.05. N.S. means not significant.

p	5	10	15	20	25	30	35	40	45	50
5		*	*	*	*	*	*	*	*	*
10			*	*	*	*	*	*	*	*
15				N.S.	*	*	*	*	*	*
20					N.S.	*	*	*	*	*
25						N.S.	*	*	*	*
30							N.S.	*	*	*
35								N.S.	N.S.	*
40									N.S.	N.S.
45										N.S.
50										

**Table 9 micromachines-13-01685-t009:** Statistical significance in dryness perception for samples with a w value of 40 µm for various p values. * Significance level < 0.05. N.S. means not significant.

p	5	10	15	20	25	30	35	40	45	50
5		*	*	*	*	*	*	*	*	*
10			*	*	*	*	*	*	*	*
15				*	*	*	*	*	*	*
20					N.S.	*	*	*	*	*
25						N.S.	*	*	*	*
30							N.S.	*	*	*
35								N.S.	*	*
40									N.S.	*
45										N.S.
50										

**Table 10 micromachines-13-01685-t010:** Statistical significance in dryness perception for samples with a w value of 50 µm for various p values. * Significance level < 0.05. N.S. means not significant.

p	5	10	15	20	25	30	35	40	45	50
5		N.S.	*	*	*	*	*	*	*	*
10			*	*	*	*	*	*	*	*
15				N.S.	*	*	*	*	*	*
20					N.S.	*	*	*	*	*
25						N.S.	*	*	*	*
30							N.S.	*	*	*
35								N.S.	*	*
40									N.S.	*
45										N.S.
50										

## Data Availability

The data presented in this study are openly available in FigShare at doi: 10.6084/m9.figshare.21071137.

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
