# Peer review of "Micromanufactured Tactile Samples for Characterization of Rough and Dry Tactile Perception"

_micromachines, 2022, doi:10.3390/mi13101685_

Round 1

Reviewer 1 Report

This is undoubtedly an excellent research manuscript that addresses a very interesting but thought-provoking question. Usually, the characterization of the roughness of the sample surface is limited to the microstructure such as scanning electron microscope and lacks the intuitive and quantitative characterization. In order to further examine the author's method of argument, I read it several times in a row. The data in the manuscript are considered to support the authors' views. As a high-quality journal, micromachines should consider publishing such excellent research papers. I therefore recommend this article for publication in micromachines without further revision.

Author Response

We cannot thank you enough for your encouraging words. 

Reviewer 2 Report

This paper presents interesting results, although the constribution is not on the side of microfabrication technologies.

The experiments are not carried out with a high number of volunteers, they are only five in some experiments.

My main concern is about the style, the paper is difficult to follow in my opinion, despite it describes a set of relatively simple experiments. The paper should be re-written to make it more readable and easier to understand.

Please add a table with the experiments you have made, their conditions and their purpose.

As an example of how difficult the reading is, you say in section 2.2 on roughness perception that the participants first rank the samples from least dry to most dry. Then "based on the obtained results", you continue with other experiments. You perform experiments about roughness and dryness perception at the same time? What is the meaning of "based on the obtained results"? Again, in the same paragraph you say "twice for each condition" but it is not clear what you refer to with "condition".

Other sentences that are not clear:

lines 38- 40

line 148

In summary, the style of the paper is not good enough and it has to be made more readable and easier to follow.

Author Response

We revised our paper following your precious comments. Please find the details of the revision in the attached file.

Round 2

Reviewer 2 Report

Dear authors,

thank you for the revised version. However, only a few changes have been made, the paper is esentially the same you sent the first time.

I suggested you added a table with your experiments, their conditions and their purpose. I would recommend a more detailed revision on your side.

Nevertheless, I will recommend the paper is accepted in its present form.

Author Response

We appreciate your comment. In the previous revision, we revised the organization of Section 2 and added some sentences about the objective of each experiment. However, as the reviewer kindly advised, we have added the table (Table 1), which summarizes the experiments we conducted in this work. We believe it makes it easy for the readers to understand how and why we conducted the experiments to deduce the conclusion of this work.
